# Effect of Isothermal Holding at 750 °C and 900 °C on Microstructure and Properties of Cast Duplex Stainless Steel Containing 24% Cr-5% Ni-2.5% Mo-2.5% Cu

**DOI:** 10.3390/ma15238569

**Published:** 2022-12-01

**Authors:** Barbara Elżbieta Kalandyk, Renata Elżbieta Zapała, Paweł Pałka

**Affiliations:** 1Faculty of Foundry Engineering, AGH University of Science and Technology, 30 Mickiewicza Ave, 30-059 Krakow, Poland; 2Faculty of Non-Ferrous Metals, AGH University of Science and Technology, 30 Mickiewicza Ave, 30-059 Krakow, Poland

**Keywords:** duplex stainless steels (DSS), microstructure, intermetallic phases, hardness, tensile test

## Abstract

Changes in the microstructure and selected mechanical properties of two-phase ferritic-austenitic cast steel containing 24% Cr-5% Ni-2.5% Mo-2.5% Cu after isothermal holding at 750 °C and 900 °C are presented. The choice of the two temperatures of isothermal holding was dictated by the precipitation of brittle phases within a range of 600 °C–950 °C, while the holding time depended on the casting cooling time in the mould. Changes in the microstructure were studied by the SEM-EDS and XRD techniques. As a result of the decomposition of the eutectoid ferrite, a σ phase that was rich in Cr, Mo, and Ni and a secondary γ_2_ austenite with Widmannstätten morphology were formed. Compared to the austenite, the chemical composition of the secondary γ_2_ austenite showed depletion of Cr and Mo. In the ferrite, the presence of Cr_2_N nitrides was also detected. After a holding time of 3 h at 900 °C, these phases increased the hardness of the tested cast steel to approximately 275 HV_10_. At the same time, the UTS value was recorded to decrease with the increasing temperature based on the tensile test results. At 750 °C, the value of UTS was 250 MPa for 1 h of holding and 345 MPa for 3 h of holding. These values decreased after increasing the temperature to 900 °C and amounted to 139 for 1 h holding and 127 MPa for 3 h holding. It was also found that the elongation values at 750 °C ranged from 7–10%, while they amounted to 35–37% at 900 °C. A fracture analysis of the tested cast steel showed that in the prevailing part, the fractures were made of ductile nature with an arrangement of dimples that is typical for this type of fracture. Non-metallic inclusions that are typical for cast steel (i.e., oxides and nitrides) were also found in the area of the fractures.

## 1. Introduction

Duplex stainless steels (DSS) are a group of modern two-phase corrosion-resistant steels that can be used under very demanding operating conditions [1,2,3], especially in extractive industries [4], petrochemicals, offshore platforms, and seawater desalination plants (the production of drinking water from seawater) [5]. The great interest in this group of alloys is due to their high corrosion resistance (among others) to chloride ions [6,7] and their good mechanical properties [8,9,10]. This favourable combination of properties is obtained by the proper content of ferrite and austenite, i.e., the ratio of ferrite-forming elements to austenite-forming elements [11,12]. For this reason, duplex steels are materials that are competitive with conventional austenitic steels (304 and 316), which are mainly used in low- and medium-aggressive environments [3]. DSS steels are also used for responsible industrial castings that operate in the mining industry [4]. The double value of the yield point and the about 30% higher tensile strength UTS of cast duplex steels also contribute to wear resistance in erosive and corrosive environments better [13]. Recently, articles have been published on the additional strengthening of the surfaces of duplex steel castings, further improving their wear resistance [14,15]. The complex chemical composition of duplex steels (especially of the third and fourth generations—referred to in the literature as Super- and Hyper-DSS [16,17]) promotes the release of undesirable and harmful secondary phases in the microstructures of these materials [18]. During their operation in corrosive environments, the presence of these phases increases the steel’s susceptibility to intercrystalline, pitting, and stress corrosion while also deteriorating its strength and welding properties [19]. The precipitation of the phases takes place within two temperature ranges, i.e., 300 °C–500 °C (α′, ε, π, G) and 600 °C–950 °C (M_7_C_3_, M_23_C_6_, Cr_2_N, CrN, σ, χ, γ’, R) [3,20]. This is the reason why these castings should not be operated at temperatures above 300 °C. The precipitates that are formed in the range of higher temperatures may lead to further deterioration of their properties and, under extreme conditions, cracks that are formed in castings with different wall thicknesses. The most dangerous is the hard, brittle and non-magnetic σ phase [21,22,23], as well as chromium carbides and nitrides. Nitrides are formed in areas with high Cr concentrations, mainly in chromium ferrite [24,25]. On the other hand, the σ phase is formed primarily at the γ/α interface. This reduces not only the mechanical properties but also the corrosion resistance [19]. The presence of Cr-enriched precipitates in the microstructure of DSS steel leads to a depletion of the matrix in Cr in their immediate vicinity, thus creating conditions for pitting or intercrystalline corrosion. These phases may occur during casting [26], welding [27,28], or plastic working [29], and the speed of their formation is related not only to the temperature but also to the time of holding at a given temperature. This phenomenon has been observed especially under the conditions of manufacturing massive castings with various cross-sections. Too long a cooling time of a casting in a mould may initiate the precipitation of harmful brittle phases. In contrast to the σ phase, chromium carbides can be almost completely eliminated from the cast microstructure by introducing elements with a higher affinity for carbon than Cr (such as Nb or Ti) during the melting. Hence, precipitates of Nb or Ti carbides, nitrides, and brittle σ phase are often found in the microstructures of such castings.

High-temperature holding of DSS steel and cast steel leads to changes in their ferritic-austenitic structure. The result is the precipitation of secondary phases, and this process depends not only on the chemical composition of steel but also on the holding time and temperature. Changes in the microstructure of DSS steel and their impact on mechanical properties have been extensively described in the literature [30,31]. The studies included the effect of time and temperature of isothermal holding on the microstructure and mechanical properties but were determined at room temperature. It was found that the extension of the isothermal holding time increased the intensity of the precipitation of secondary phases (including σ phase, chromium nitrides and carbides, and γ_2_ phase), which, in turn, increased the strength of DSS steel and reduced its plastic properties [18]. It is known that, at high temperatures, the character of changes in strength and plastic properties is different [32]. Considering the importance of the impact of the tensile test temperature on the properties of cast DSS steel, the authors decided to perform strength tests at a temperature in the range of secondary phase precipitation, i.e., 600 °C–950 °C. The purpose of this article was to link selected mechanical properties obtained after the high-temperature tensile test carried out at 750 °C and 900 °C with the microstructure of the tested cast 24% Cr-5% Ni-2.5% Mo-2.5% Cu steel after isothermal holding at 750 °C and 900 °C for 1 h and 3 h. Moreover, it was the authors’ intention to show how the temperature and the time of isothermal holding affect changes in the microstructure and, thus, changes in the mechanical properties of the tested cast steel. This may prove to be important when the crack formation is studied in small intricate castings that are made of the tested material during their cooling in a mould.

## 2. Materials and Methods

Samples of cast duplex steel with the chemical composition that is presented in Table 1 were used in the tests. The PREN (Pitting Resistance Equivalent Number) is an indicator of the resistance of DSS steel to pitting corrosion and depends on the chemical composition of the steel, the value of the molten cast steel (calculated from the PREN formula: PREN = %Cr + 3.3%Mo + 16%N), was 33.25 [33]. It shows the significant influence of Cr, Mo, and N on the resistance to this type of corrosion. Additionally, it is used in the classification of this group of alloys. The cast steel was melted in a laboratory induction furnace with a 30 kg capacity. The charge was composed of Armco iron, Cr-Ni steel scrap, pure elements of Cr, Mn, Ni, Mo, and Cu, and ferroalloys such as FeSi and FeNb. The bath was deoxygenated with Al and CaSi. Because the carbon content in the melt was greater than 0.02%, Nb was introduced to produce Nb carbides and/or carbonitrides in place of Cr carbides. The “Y” type test ingots were subjected to heat treatment (including solution annealing from 1080 °C) in order to provide a two-phase ferritic-austenitic structure that was free from secondary phases. Then, the tested material was held at two temperatures that fell within the range of the occurrence of brittle phases, i.e., 600 °C–950 °C. Temperatures of 750 °C and 900 °C were selected for the tests. The holding times were 1 and 3 h at each temperature.

Microstructure examinations were carried out on a Leica MEF4M light microscope (made in Wetzlar, Germany) at magnifications of 100× through 1000× and on a JEOL JSM 7100 F secondary electron (SE) field emission scanning electron microscope (SEM) (made in Tokyo, Japan) at magnifications of 100× through 7500×. The chemical composition of the microstructural constituents in the microregions was determined with an EDS detector from Oxford Instruments (made in Abingdon, UK). The examinations were carried out on specimens that were etched with Beraha’s reagent (0.3 g K_2_S_2_O_5_, 20 mL HCl, and 80 mL H_2_O). For the SEM-EDS and XRD studies, samples after deep etching in a mixture of hydrochloric and nitric acids were also used. Qualitative and quantitative examinations of the non-metallic inclusions that were present in the tested cast steel were also carried out. The examinations were carried out on non-etched samples, which served for the evaluation of the microstructure after the etching. ImageJ software was used to calculate the number of non-metallic inclusions. For each sample, the total area of the examined polished sections was about 2 mm^2^. The ferrite content in the examined samples was measured with a Fischer FMP30 feritscope (made in Sindelfingen, Germany). The content of the ferromagnetic phase (ferrite) was determined by the method of magnetic induction using a Fischer FMP30 Feritscope. Measurements of ferrite content made with this device comply with the ISO 17655 standard. Using this device, the ferrite content measuring range was 0.1–80%. The ferrite fraction was determined in five areas for each of the tested samples. Hardness was determined on metallographic specimens by the Vickers method (HV_10_ and HV_0.02_) using a Nexus 4000 hardness tester from Innovatest (made in Maastricht, The Netherlands) [34]. The X-ray qualitative phase analysis was performed on a Siemens D500 diffractometer (made in Munich, Germany). The parameters of the X-ray diffraction studies are given in Table 2. At 750 °C and 900 °C, a uniaxial tensile test was carried out in accordance with PN-EN 6892-1: 2016-09 [35] using round samples with a diameter of 3 mm and a datum base length of 24 mm. The tests were carried out at a constant strain rate of ε = 10^−3^ [1/s] on an Instron TM5 (made in Norwood, MA, USA) testing machine that was equipped with a measuring head of 5 kN as well as a resistance furnace with temperature stabilisation and with an additional system that monitored the sample temperature during the stretching.

## 3. Results and Discussion

### 3.1. Non-Metallic Inclusions in Tested Cast Steel

Compared to the arc furnace melting process, the technology of cast steel melting in an induction furnace is characterised by an increased content of non-metallic inclusions in the final cast products. The presence of inclusions may adversely affect the properties of the cast steel; therefore, it was decided to determine the content of these inclusions. The results of the calculations of the content of the non-inclusions in the examined cast steel samples (Figure 1) showed similar contents that ranged between 1.02–1.06% for those samples that were held at 750 °C and between 1.03–1.19% for those samples that were held at 900 °C. As for the technology of cast steel melting in an induction furnace, the obtained values were within acceptable limits. Examinations of the cast steel that were carried out by light microscopy showed the presence of single polygonal inclusions that corresponded to the nitrides. These were evenly distributed on the surface and characterised by an irregular shape.

### 3.2. Microstructure of Tested Cast Steel

Before the isothermal holding, the cast duplex steel was characterised by a two-phase microstructure that was composed of ferrite and austenite. Examples of the microstructure of the examined cast steel after isothermal holding at 750 °C and 900 °C for 1 and 3 h are shown in Figure 2 and Figure 3. The microstructure examinations that were carried out by light microscopy at 100× and 500× magnifications on the cast steel samples that were held at 750 °C for 1 h did not reveal the presence of secondary phase precipitates; these were only found in the cast steel microstructure after extending the holding time to 3 h (Figure 2d and Figure 3c,d). Increasing the temperature to 900 °C accelerated the precipitation process of the secondary phases. After both 1 and 3 h, these phases were noticed to appear at the γ/α interface (Figure 2d and Figure 3d). At both temperatures and both times of holding, the cast steel microstructure showed the presence of γ_2_ austenite in the form of fine coniferous precipitates that were similar to the Widmanstätten structure. The presence of the σ phase and γ_2_ austenite was associated with the eutectoid decomposition of the ferrite into the σ phase and secondary austenite (γ_2_) [36]. The examinations that were carried out with a ferritoscope confirmed the drop in ferrite content along with the extension of the holding times at both temperatures (Figure 4); thus, it was shown that extending the holding time from 1 to 3 h was the main factor that reduced the content of the magnetic phase (i.e., ferrite, at 750 °C and 900 °C) and promoted the formation of non-magnetic σ phase [37]. The obtained results demonstrated that in the manufacture of DSS steel castings, it is necessary to accelerate the casting cooling process in the mould.

The reaction of the ferrite decomposition into the σ phase and secondary austenite (γ_2_) was additionally confirmed by the examinations of the microstructure that used scanning microscopy. The examinations confirmed the presence in the microstructure of this phase, which occurred mainly at the γ/α boundary. Due to the limited possibilities of light microscopy, the examinations of the microstructure of the tested cast steel were extended and additionally included SEM and XRD. The SEM-EDS studies showed high contents of Cr (approx. 30–32%), Ni (approx. 4–5%), and Mo (approx. 5–7%) in the examined precipitates that occurred at 750 °C for 1 and 3 h (Figure 5 and Figure 6). The distribution of the molybdenum in the examined precipitates that were located at the γ/α interface is shown in Figure 6. The high contents of Mo and Cr in relation to the matrix indicate the presence of the σ phase; this was further confirmed by X-ray diffraction analysis (Figure 7). The SEM-EDS studies also showed that the precipitates of the σ phase that were present in the sample that was held for 1 h did not form a continuous network. On the other hand, holding for 3 h contributed to the growth of this phase (which assumed the form of continuous precipitates at the γ/α interface, growing in the direction of the Cr- and Mo-rich ferrite—Figure 5b and Figure 6).

The precipitates of the σ phase were also observed at 900 °C for 1 and 3 h (Figure 8). The chemical composition of the phase after holding at 750 °C and 900 °C was comparable in terms of the main constituents. Additionally, the SEM-EDS and XRD studies revealed the presence of Cr_2_N nitrides that accompanied this phase (Figure 9); this is very likely due to the content of 0.04% N in the tested cast steel. Chromium nitrides can be released within a range of 600° through 900 °C inside the ferrite grains and at the ferrite/ferrite interface [38]. Their formation is accelerated by the presence of Cr, Mo, and W, which also increase their stability at high temperatures.

A comparison of the obtained XRD diffraction records after traditional and deep etching indicates the etching of the ferritic phase and (along with this phase) fine Cr_2_N nitrides (Figure 7 and Figure 9). Additionally, the use of deep etching on the cast steel allowed for confirming the presence of the secondary austenite (γ_2_) with the Widmanstätten morphology in the SEM-EDS studies (Figure 10). The SEM-EDS analysis of the chemical composition of the austenite and secondary austenite (γ_2_) showed that the secondary austenite contained lower content of Cr and Mo and a slightly higher content of Ni (Table 3) [37].

The SEM-EDS studies also showed that in the microstructure of the examined cast steel (at both 750 °C and 900 °C), precipitates in the form of fine needles were present at the γ/α interface and in the ferritic phase. The analysis of the chemical composition revealed extended enrichment in the Nb (approx. 34–49%), Cr (12%), and N (2.1%) (Figure 11). This may indicate the presence of the complex non-equilibrium nitrides that have been described in the literature [16]. Their formation was triggered by both the technological process (the deliberate introduction of Nb) and the supersaturation of ferrite with nitrogen during the cooling of the tested cast steel. Due to the low chromium content in the examined precipitates (as compared to the Cr_2_N or CrN), their impact on the occurrence of pitting corrosion will be rather moderate. The XRD examinations did not show the presence of M_23_C_6_ chromium carbides; this means that the addition of the Nb that was introduced to the examined cast steel effectively blocked their release in an amount that did not exceed the detectability threshold.

### 3.3. Effect of Isothermal Holding on Hardness of Tested Cast Steel

Hardness measurements showed an increase in the hardness with the longer time and higher temperature of the isothermal holding. The presence of the secondary phases in the microstructure of the examined cast steel also contributed to the recorded increase in the hardness (Table 4). According to [39], the occurrence of the σ phase can serve as a means of increasing the abrasion resistance of castings that are made from this grade of steel to a certain extent. The measured microhardness of the σ phase was within a range of 560–672 HV_0.02_. Large differences in the obtained values of the microhardness resulted from the different sizes of this phase and the depth of its embedding in the matrix (among others things).

### 3.4. Tensile Strength of Tested Cast Steel at 750 °C and 900 °C

Examples of the tensile curves that were recorded for the tested cast steel at 750 °C and 900 °C are shown in Figure 12. Before the static tensile test, the tested cast steel was subjected to solution treatment at 1080 °C and isothermal holding at 750 °C and 900 °C in two time variants, i.e., 1 h and 3 h.

The results of the tensile test showed that the stresses were higher in the tested cast steel at 750 °C as compared to 900 °C. At the temperature of 750 °C, the tensile strength values were 250 for the 1 h treatment and 345 MPa for the 3 h treatment. Extending the holding time at 750 °C from 1 h to 3 h increased the yield strength and tensile strength. At the isothermal holding temperature of 750 °C (the initial range of temperatures of secondary phase precipitation for DSS steel, i.e., 650 °C–950 °C), the extension of the isothermal holding time from 1 h to 3 h had a significant impact on the observed microstructure (Figure 2 and Figure 3). This applies in particular to the reduced ferrite content in the structure (Figure 4) and the appearance of the sigma phase at the γ/α interface. After isothermal holding for 3 h, this phase took the form of thin, continuous precipitates growing towards ferrite, as shown in Figure 5b and Figure 6.

The increase in strength properties at a temperature of 750 °C along with the extension of the holding time, may be due to the increasing content of the precipitated σ phase, which in small amounts can control the strengthening process of the tested cast steel (Figure 13c and Figure 14a). On the other hand, the elongation values obtained for these materials during the high-temperature tensile test at 750 °C decreased with the increasing temperature and amounted to approximately 9 and 7% in the case of samples held at 750 °C for 1 h and 3 h, respectively. This generally corresponds to changes in the mechanical properties of DSS steel subjected to a static tensile test at room temperature [40].

Another series of tensile tests were carried out at 900 °C. The strength values obtained for cast steel in the tensile test at 900 °C were significantly lower than those obtained for cast steel in the tensile test at 750 °C. On the other hand, a significant increase in elongation was observed (Figure 12).

At the temperature of 900 °C, for both holding times, the curves differed only slightly with the values of 127 (3 h) and 139 MPa (1 h). The elongation at 900 °C was higher and comprised within a range of 35–37%. Due to the limited number of samples used in the tests, only at a temperature of 900 °C, it could be concluded that the holding time of 1 or 3 h had no substantial effect on the stress values, contrary to 750 °C, where a longer holding time increased the stress by nearly 100 MPa. In none of the time variants, the cast steel after isothermal holding at 900 °C showed such a clear growth of the σ phase as was observed after isothermal holding at 750 °C (Figure 2, Figure 3 and Figure 8). Precipitates of this phase were larger and had a more compact morphology than after isothermal holding at 750 °C (Figure 3 and Figure 8), which explains the minor role of isothermal holding time in shaping the mechanical properties obtained in the tensile test carried out at 900 °C (Figure 13d and Figure 14b).

The UTS and YS values obtained at 750 °C and 900 °C were much lower than the UTS and YS values obtained for the cast steel tested at room temperature (UTS: 720–770 MPa, YS: 600–620 MPa) [13]. Thus, the obtained results confirm the literature data, which claims that with the increase in test temperature, the strength values decrease and the plastic properties increase [32]. The increase in the temperature of the static tensile test and the presence of numerous precipitates of secondary phases in the microstructure of the tested cast steel contributed to the reduction of strength properties.

The results obtained in the static tensile test at elevated temperatures are confirmed by the microrelief (character) of fractures.

### 3.5. Fracture Analysis

The temperature of the tensile test was found to have a significant impact on the microrelief of the fracture surface (Figure 13 and Figure 14). Reheating the tested cast steel to a high temperature during the tensile test activated the diffusion process, resulting in the growth of the existing grains (Figure 13c,d,g,h and Figure 14a,b,e,f) and in the nucleation of new precipitates of secondary austenite (Figure 13c,d and Figure 14a,b) [41]. These precipitates improve ductility but reduce corrosion resistance. The nucleation of secondary austenite precipitates in the ferritic matrix of the tested cast steel is related to the earlier precipitation of Cr nitrides (Figure 7 and Figure 9) [28]. The SEM analysis revealed that, at the temperature of 750 °C, the ductile fracture prevailed regardless of the cast steel’s holding time. This was characterised by a typical arrangement of dimples and micro-dimples (with precipitates visible inside them—Figure 13e,g and Figure 14c,e). The extension of the time from 1 to 3 h at 750 °C resulted in the growth of the precipitates that could be observed in the fracture. A similar situation occurred when the fractures that were obtained at 750 °C were compared to those that were obtained at 900 °C. The central part of each fracture (in the axis of the sample) was subjected to detailed examinations. Macro examinations showed varied topographies of the surfaces of the examined fractures. In the case of cast steel subjected to the tensile test carried out at 750 °C, the occurrence of voids characteristic of the ductile fracture mechanism was observed along with changes in the planes of crack front movement (Figure 13e,g and Figure 14c,e). In the case of cast steel subjected to the tensile test carried out at 900 °C, the fracture surface was not clear enough to determine which fracture mechanism was prevailing. The reason was a large number of finely dispersed precipitates (Figure 13f,h and Figure 14f).

In fractures obtained from the tensile test carried out at 750 °C, the ductile mode of fracture prevailed. It was accompanied by the presence of characteristic dimples formed around the precipitates; a brittle fracture mechanism was also observed to occur locally. On the other hand, in fractures obtained from the tensile test carried out at 900 °C, the sigma phase present in large amounts (Figure 13d and Figure 14b) did not allow for unambiguous identification of the dominant fracture mechanism. In Figure 13f, voids characteristic of the ductile fracture mechanism were observed, but in Figure 13h, the characteristic dimples, similar to those present in cast steel subjected to the tensile test carried out at 750 °C, did not appear (Figure 13g). The same results were obtained for the longer time of isothermal holding, which means that in cast steel subjected to the tensile test carried out at 750 °C, the fracture could still be clearly identified as ductile. On the other hand, after the tensile test carried out at 900 °C, the density of the emerging phases made an unambiguous characterisation of the obtained fractures very difficult, although a fragment of the ductile fracture surface was observed (Figure 14f).

Additionally, the identification of particles present in the fracture was carried out (Figure 15). Large particles enriched in N, O, and Al were observed; they fractured in a brittle mode, and, at the same time, they initiated the brittle fracture process in their close proximity. Because these particles were not dominant inclusions in the tested cast steel, they did not play a significant role in the cast steel cracking process.

In addition to the non-metallic inclusions revealed on fractures, the presence of the σ phase in the microstructure of the examined cast steel (Figure 13c,d and Figure 14a,b—microstructure obtained in the heads of tensile samples) can be a source of the high concentration of tensile stresses, which in combination with thermal stresses can lead to crack formation in castings. This phenomenon may occur especially during the cooling of castings. Therefore, in the process of casting DSS steel into foundry moulds, it is very important to remember that holding castings in the range of 750 °C–900 °C for even 1 h will favour the formation of a brittle and hard σ phase. The conducted research has shown that this is a real threat, especially in the production of medium and large castings (>0.5 Mg) with varied wall thicknesses [39]. To avoid adverse consequences, castings should be knocked out from moulds before they enter the range of secondary phase precipitation, i.e., before they reach 950 °C.

## 4. Conclusions

Microstructure examinations and tensile tests of cast duplex steel led to the following conclusions: As a result of isothermal holding at 750 °C and 900 °C for the 1 and 3 h, the following secondary phases precipitated in the examined cast steel: σ phase, γ_2_ secondary austenite with Widmanstätten morphology and Cr_2_N nitrides. These phases contributed to the increased hardness of the tested cast steel. The σ phase present in the microstructure of the examined cast steel contains approx. 30–33% Cr, 3–5% Ni and 5–7% Mo and is characterised by a diverse morphology depending on the temperature and time of holding. In the cast steel held for 3 h at both temperatures, it tended to form continuous precipitates at the γ/α interface.In the performed static tensile test, higher stress values were obtained at 750 °C than at 900 °C. The tensile strength values were 250 and 345 MPa for the 1 and 3 h treatments, respectively, while the elongation was comprised in a range of 7–10%. At the temperature of 900 °C, the tensile strength decreased significantly and amounted to 139 and 127 MPa for the 1 and 3 h treatments, respectively, while the elongation increased to 35–37%.SEM studies of the fractures showed that, regardless of the applied temperature of the isothermal holding, the microrelief of the fracture surfaces was mainly ductile. Additionally, the increase in the temperature to 900 °C increased the dimensions of the precipitates that could be observed in the fractures. The conducted studies also revealed the presence of non-metallic inclusions on their surface.

## Figures and Tables

**Figure 1 materials-15-08569-f001:**
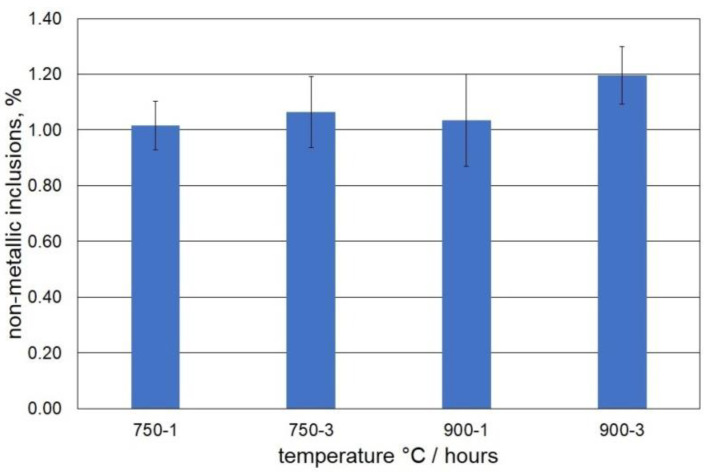
Non-metallic inclusions in the tested cast steel.

**Figure 2 materials-15-08569-f002:**
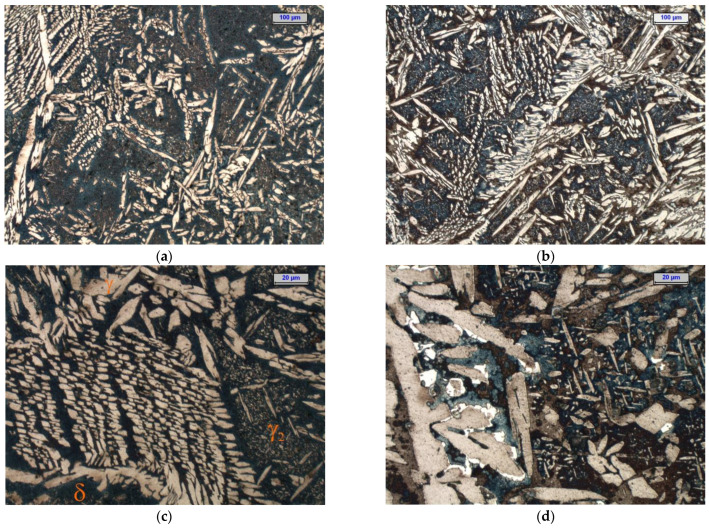
Microstructure of tested cast steel after isothermal holding for 1 h at 750 °C (**a**,**c**) and 900 °C (**b**,**d**)—light microscope, etching with Beraha’s reagent (dark—ferrite; beige—austenite; white—σ phase and carbides); (**a**,**b**)—100× magnification and (**c**,**d**)—500× magnification.

**Figure 3 materials-15-08569-f003:**
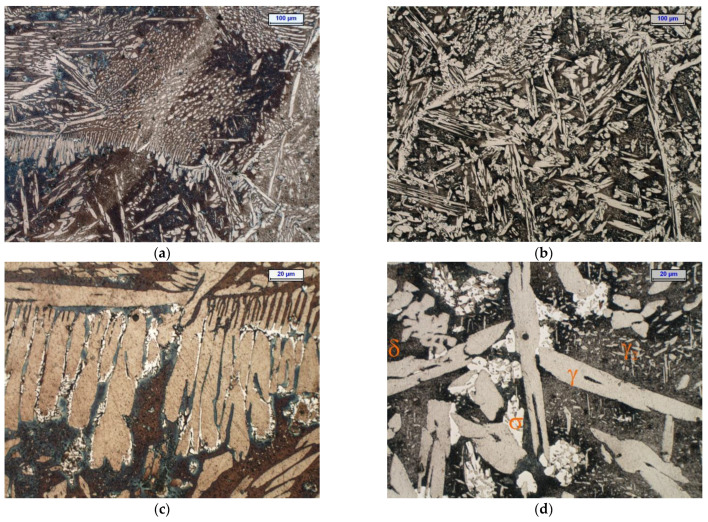
Microstructure of tested cast steel after isothermal holding for 3 h at 750 °C (**a**,**c**) and 900 °C (**b**,**d**)—light microscope, etching with Beraha’s reagent (dark—ferrite; beige—austenite; white—σ phase and carbides); (**a**,**b**)—100× magnification and (**c**,**d**)—500× magnification.

**Figure 4 materials-15-08569-f004:**
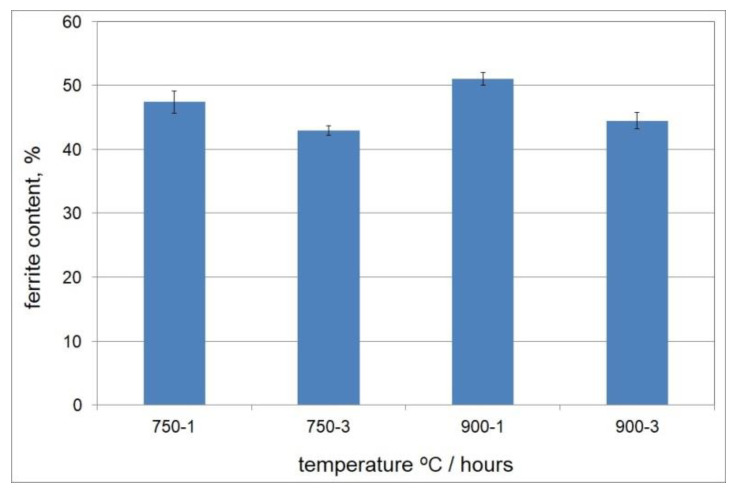
Effect of holding time and temperature on ferrite content in tested cast steel.

**Figure 5 materials-15-08569-f005:**
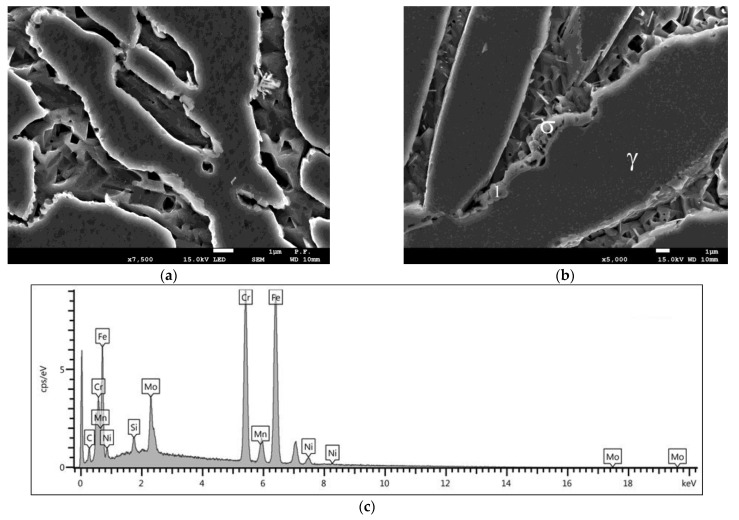
SEM image of microstructure of tested cast steel after holding at 750 °C for 1 h (**a**) and 3 h (**b**) (deep etching); (**c**) example of spectrum of precipitate occurring at γ/α interface (Point 1, **b**); (scanning microscope).

**Figure 6 materials-15-08569-f006:**
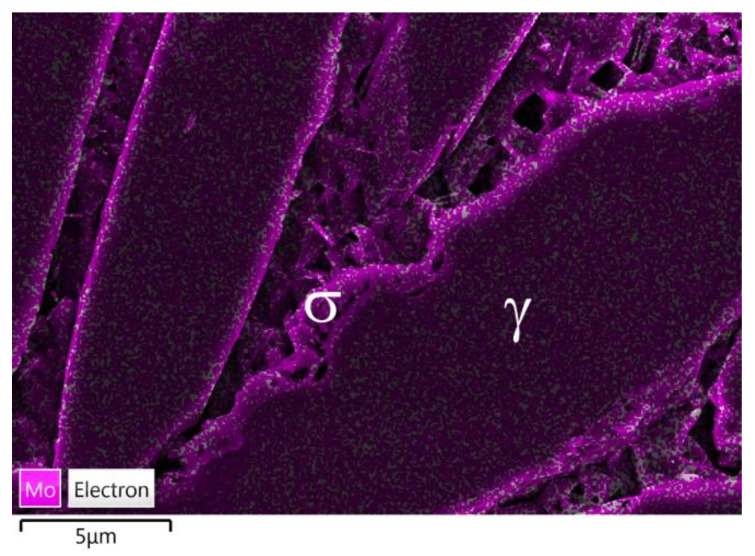
Tested cast steel held at 750 °C for 3 h—Mo distribution in precipitate shown in Figure 5b.

**Figure 7 materials-15-08569-f007:**
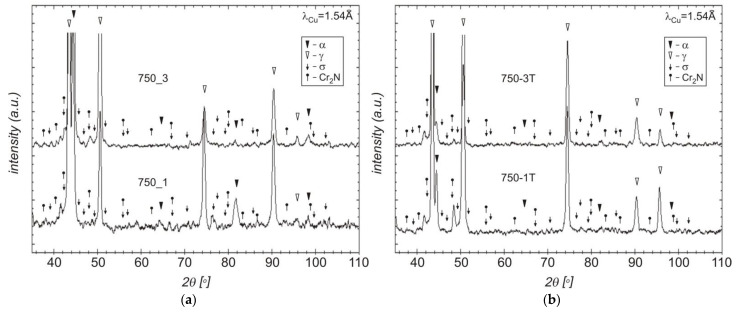
Diffractogram of tested cast steel after holding at 750 °C for 1 and 3 h: (**a**) traditional etching; (**b**) deep etching (T in denotations).

**Figure 8 materials-15-08569-f008:**
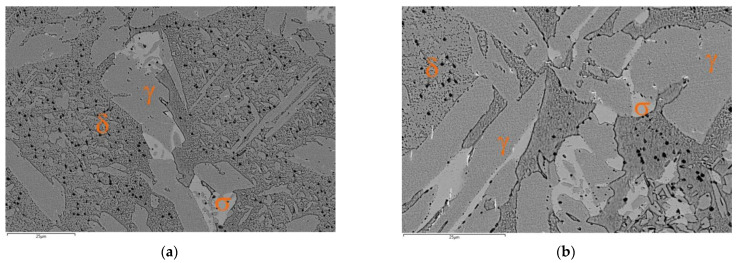
SEM image of tested cast steel after isothermal holding at 900 °C for (**a**) 1 h and (**b**) 3 h.

**Figure 9 materials-15-08569-f009:**
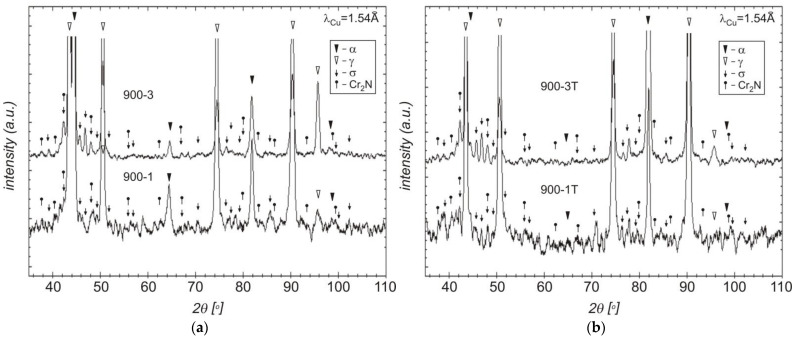
Diffractogram of tested cast steel after holding at 900 °C for 1 and 3 h: (**a**) traditional etching; (**b**) deep etching (T in denotations).

**Figure 10 materials-15-08569-f010:**
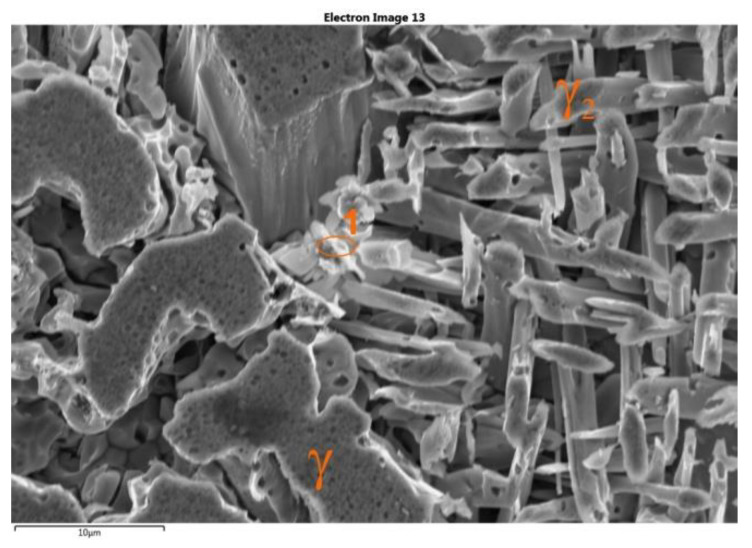
Example of precipitates of γ and γ_2_ phases in microstructure of tested cast steel after deep etching (at 900 °C for 3 h) (scanning microscope).

**Figure 11 materials-15-08569-f011:**
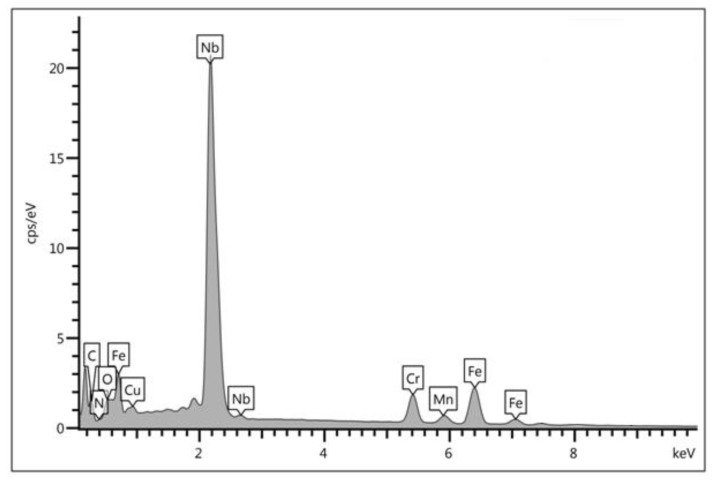
EDS X-ray spectrum of precipitate 1 from Figure 10.

**Figure 12 materials-15-08569-f012:**
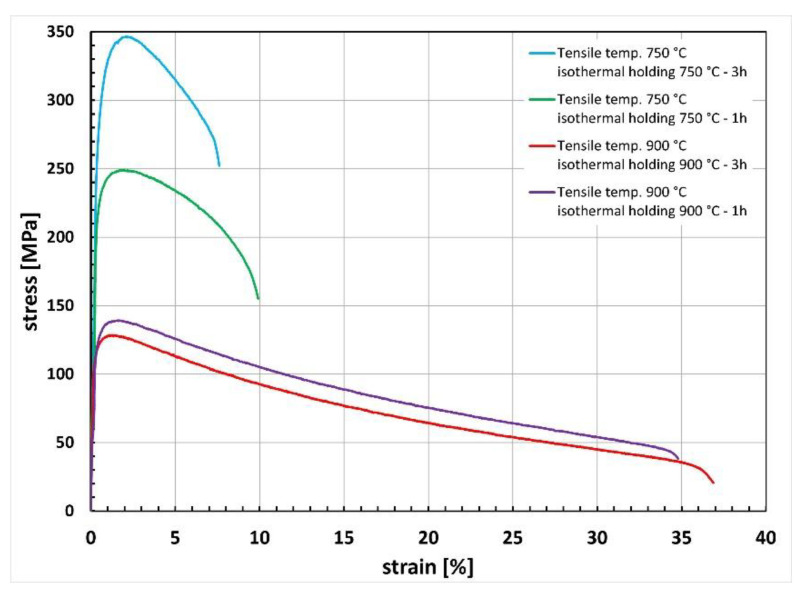
Characteristics of the stress–strain curves obtained at 750 °C and 900 °C for samples after solution heat treatment at 1080 °C and isothermal holding at 750 °C and 900 °C for 1 h and 3 h.

**Figure 13 materials-15-08569-f013:**
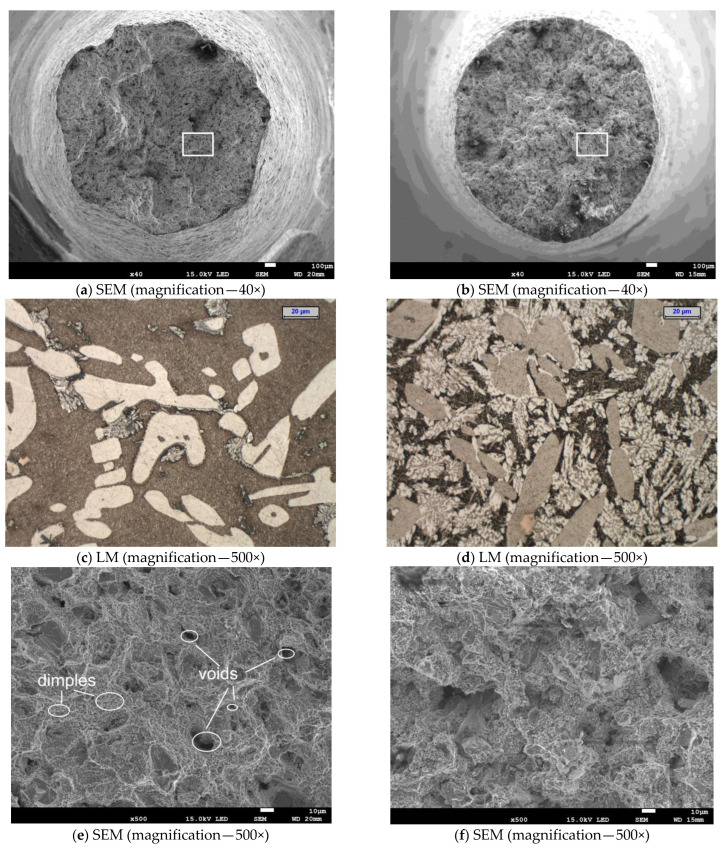
SEM image of fracture surface after tensile test (at 750 °C and 900 °C) and prior isothermal holding at 750 °C for 1 h (**a**,**e**,**g**) and at 900 °C for 1 h (**b**,**f**,**h**); LM microstructure from tensile specimen (**c**,**d**); images (**e**,**f**,**g**,**h**) from the frame area.

**Figure 14 materials-15-08569-f014:**
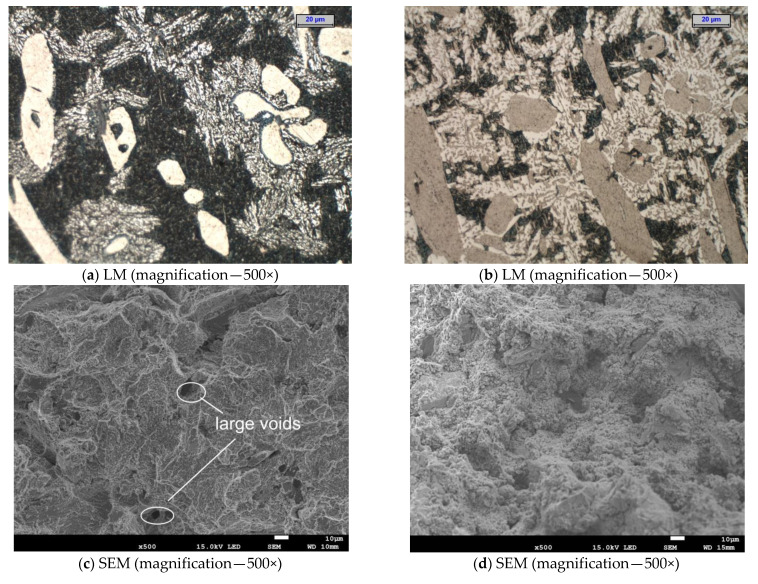
SEM image of fracture surface after tensile test (at 750 °C and 900 °C) and prior isothermal holding at 750 °C for 3 h (**c**,**e**) and at 900 °C for 3 h (**d**,**f**); LM microstructure from the tensile specimen (**a**,**b**).

**Figure 15 materials-15-08569-f015:**
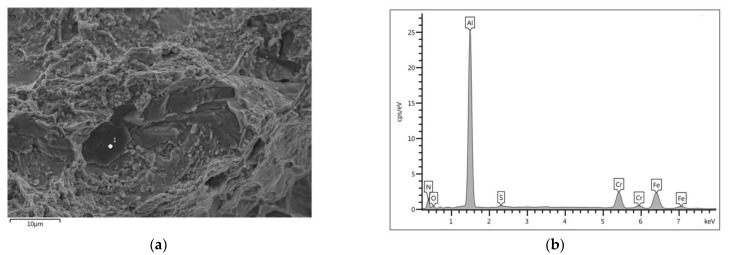
SEM image of cast steel fracture after tensile test and prior isothermal holding at 750 °C for 1 h (**a**)—with X-ray spectrum of the AlN area (Point 1 on microstructure) (**b**).

**Table 1 materials-15-08569-t001:** Chemical composition of tested cast steel.

**Melt**	C	Cr	Ni	Mo	Mn	Si	Cu	P	S	Nb	N
	wt. %										
**DSS**	0.06	24.2	5.2	2.55	1.0	0.4	2.7	0.01	0.01	0.25	0.04

**Table 2 materials-15-08569-t002:** Parameters of X-ray diffraction.

Parameters of X-ray Diffraction	Tested Cast Steel
Angular range, deg	2ϴ 35–110 deg
Step size, deg	0.04
X-ray tube	CuKα radiation, λ_Cu_ = 1.54 Å
Voltage, kV	40
Current, mA	35

**Table 3 materials-15-08569-t003:** Chemical composition at examined points—Figure 10.

Spectrum Label	Si	Cr	Mn	Ni	Cu	Mo	Fe
	wt. %						
γ from Figure 10	0.5	21.4	1.0	7.2	3.7	1.9	Bal
γ_2_ from Figure 10	0.5	18.6	1.4	8.2	3.5	1.6	Bal
γ at 750 °C *	0.4	21.3	1.1	6.9	3.9	1.8	Bal
γ at 900 °C *	0.4	21.6	1.15	6.9	3.7	1.8	Bal

* / examinations were carried out on polished metallographic sections.

**Table 4 materials-15-08569-t004:** Hardness of examined cast steel (averages of five measurements) HV_10_.

	Temperature and Time of Holding [°C/h]			
	750-1	750-3	900-1	900-3
Hardness and standard deviation	234 ± 0.58	247 ± 1.0	253 ± 1.52	275 ± 1.15

## Data Availability

Data sharing not applicable, all the data created for this study are already displayed in the article.

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
