# Peer review of "Effect of Isothermal Holding at 750 °C and 900 °C on Microstructure and Properties of Cast Duplex Stainless Steel Containing 24% Cr-5% Ni-2.5% Mo-2.5% Cu"

_materials, 2022, doi:10.3390/ma15238569_

Round 1
Reviewer 1 Report
The overall manuscript and results are exciting, and the methodology is reasonable. The text was written quite well, and only a few corrections are necessary. However, The authors should address the following issues with the manuscript before the paper can be considered for publication.
Please consider the following comments and open questions.
1- High-temperature tensile properties of the duplex stainless steel with similar chemistry as the studied alloy have been investigated before, e.g., Materials Characterization 58 (2007) 65–71 and Materials Characterization 59 (2008) 1776 – 1783. The influence of secondary sigma phase precipitation and its morphology depending on the aging temperature and time have been well established in several papers. Therefore, the authors should highlight the novelty of their work in the paper and explain how their findings add to the existing literature results.
2- The authors mentioned that the samples held at 750°C for 1 h did not show the presence of secondary phase precipitations, as concluded from Fig. 2. However, XRD shows sigma phase reflections, contrary to the statement. A discussion is required here.
3- I suggest showing the microstructure of the un-aged sample.
4- Did the authors detect Chi, χ, particles at the austenite/ferrite boundary in the aged samples?
5- In Table 3, what is srednia?
6- As shown in Materials Characterization 59 (2008) 1776 – 1783, the strength and elongation of duplex stainless steel show a slight variation with isochronal ageing temperature (600-900°C). In contrast, the present paper exhibits a significant change in strength and ductility as ageing temperature increases. This discrepancy should be explained and justified.
7- The authors mentioned that with increasing ageing temperature to 900°C, the kinetics of sigma precipitation accelerates, which is also visible from the microstructure images. Such an increase in sigma phase fraction with temperature is expected to increase, or at least not reduce, the strength at 900°C compared to 750°C. However, Fig. 12 shows a remarkable reduction of strength at 900°C. A detailed discussion is required here regarding the volume fraction of sigma precipitates and the possible effect of microstructure coarsening, particularly the sigma phase, on strength.
8- I wonder whether cracks formed in 900°C/3h samples after deformation at the γ/σ interface due to a continuous network of precipitates?
9- Considering that the sigma phase forms from the decomposition of ferrite, can one expect a lower ferrite content at 900°C because of a higher fraction of the sigma phase? If yes, why Figure 4 shows higher ferrite content at 900°C than at 750°C?
Reviewer 2 Report
The following points are necessary to address clearly for further processing;
[1]. Remove some special characters from the abstract in lines 7 and 8.
[2]. Create harmony in-between the abstract and the last paragraph of the introduction, like the annealing word is not utilized in the abstract.
[3]. What is PREN in heading 2? Write the abbreviation first.
[4]. Explain the importance of calculating the PREN value for readers.
[5]. What is this in heading 3.1., 1.02 ÷ 1.06%, 1.03 ÷ 1.19%?
[6]. For heading 3.2, just writing the microstructure as a heading seems quite simple. Modify the heading title. Same for heading 3.3.
[7]. In Fig. 2 on Page 5, writing 750-1 and 900-1 is not appropriate. Draw a text box and write inside the Figure or mention it in the caption. Same for the other Figures as well.
[8]. How you have analyzed the ferrite content percentage in Fig. 4, the same for Fig. 1?
[9]. The explanation for heading 3.4 is quite simple and not enough. Relate this section to the previous section of microstructure.
[10]. The explanation of Figures 13, 14, and 15 is completely missing. Add enough argument and explanation for heading 3.5.
[11]. Why microstructures are added along with the fracture surface in Fig. 13, 14, and 15?
[12]. The first point of the conclusion needs to be written again.
Round 2
Reviewer 1 Report
The authors sufficiently addressed the issues raised by me. However, I think the Introduction is still missing the novelty of the work and explaining how the findings add to the existing literature results.
Author Response
Thank you very much for your valuable comments on our article.
Please see the attachment.

Reviewer 2 Report
Don’t know why the symbol of divide (÷) is introduced in the abstract and other places as well.
For point 4, I didn’t find the answer in the revised manuscript: how can we calculate the PERN, with what method or equipment, or how you have calculated it?
For Point 5, still, the query is not resolved, is it the standard to show the range with the ÷ symbol?
For Point 9, highlight only the new written section. This section is still weak.
For Point 10, the explanation is not enough and synchronizes. Need to arrange again with enough arguments and explanations.
Highlight the region in Fig. 13(a,b), where the other magnified micro fracture surface was captured as Fig. 13(c,d,e,f).
Properly explain Fig .15 along with the X-ray spectrum.
Author Response

(The authors gave the same response as above.)
